# Genetic Parameters Estimation of Milking Traits in Polish Holstein-Friesians Based on Automatic Milking System Data

**DOI:** 10.3390/ani11071943

**Published:** 2021-06-29

**Authors:** Joanna Aerts, Dariusz Piwczyński, Heydar Ghiasi, Beata Sitkowska, Magdalena Kolenda, Hasan Önder

**Affiliations:** 1Lely Services B.V., Cornelis van der Lelylaan 1, 3147 PB Maassluis, The Netherlands; JAerts@lely.com; 2Department of Animal Biotechnology and Genetics, Faculty of Animal Breeding and Biology, UTP University of Science and Technology, 85-084 Bydgoszcz, Poland; beatas@utp.edu.pl (B.S.); kolenda@utp.edu.pl (M.K.); 3Department of Animal Science, Faculty of Agricultural Science, Payame Noor University, Tehran P.O. Box 19395-3697, Iran; ghiasei@gmail.com; 4Department of Animal Science, Ondokuz Mayis University, Samsun 55139, Turkey; honder@omu.edu.tr

**Keywords:** milk, Holstein-Friesian, milking robots, heritability, correlations

## Abstract

**Simple Summary:**

The automatic milking system provides a large amount of information characterizing the course of each cow milking, which is not available in the conventional system. From a breeding point of view, it is interesting to establish the genetic variability of these traits, as well as to establish the relationship between them. The aim of our study was to estimate heritability and genetic correlations for milk yield (MY), milking frequency (MF), and speed (MS) for 1713 Polish Holstein-Friesian primiparous cows milked in barns with an automatic milking system. Our study conclusively indicated that it is possible to carry out effective selection for milking speed, which provides an opportunity to increase the number of cows per milking robot, and thus increase the profitability of production in the herd. We proved that selection for milk yield and daily milking frequency is also feasible. Our research showed a high, positive genetic correlation between milking frequency and milk yield, which allows us to conclude that preferring breeding cows with a natural tendency to frequent visits to the milking robot should indirectly improve the genetic basis of milking.

**Abstract:**

The automatic milking system (AMS) provides a large amount of information characterizing the course of each milking cow, which is not available in the conventional system. The aim of our study was to estimate heritability and genetic correlations for milk yield (MY), milking frequency (MF), and speed (MS) for 1713 Polish Holstein-Friesian primiparous cows milked in barns with an AMS. Daily heritability indicators estimated using second-order Legendre polynomials and Random Regression Models showed high variation during lactation, ranging 0.131–0.345 for MY, 0.153–0.322 for MF, and 0.336–0.493 for MS. The rates of genetic correlation between traits ranged: 0.561–0.929 for MY-MF, (−0.255)−0.090 for MF-MS, (−0.174)−0.020 for MY-MS. It is possible to carry out effective selection for milking speed, which provides an opportunity to increase the number of cows per milking robot, and thus increase the profitability of production in the herd. The results proved that selection for milk yield and daily milking frequency is also feasible. The research showed a high, positive genetic correlation between milking frequency and milk yield, which allows us to conclude that preferring breeding cows with a natural tendency to frequent visits to the milking robot should indirectly improve the genetic basis of milking.

## 1. Introduction

Genetic parameters can be estimated from varying amounts of information concerning milk yield traits of evaluated animals and their relatives—in the simplest case, based on lactation milk yield or milk yield standardized to 305 lactation days [1,2,3]. The current standard for estimating the genetic parameters in most countries is to use test-day (TD) yields recorded regularly as part of cow milk recording schemes [4,5,6,7,8]. Test-day models (TDM) allow for modeling daily milk yield based on changing environmental conditions on different days and stages of lactation [9]. An important issue related to the estimation of the genetic parameters, which has been the subject of research since the 1960s, is the choice of a function describing the shape of the lactation curve. Many different proposals for lactation curve modeling have been presented, including the functions, among other of Ali and Schaffer [10], Wilmink [11], and, since 1994, Legendre orthogonal polynomials [12], which produce varied results in terms of lactation curve modeling quality [13]. Functions of the lactation curve can be included in the calculations as fixed regression models or random regression models (RRM) [6,8,9,14,15]. The first model assumes that the effects of animal and fixed environmental factors are identical on each day of lactation, while the latter assumes they are different [16]. RRM enables genetic parameters to be estimated for the whole lactation and successive days of lactation, and furthermore the lactation curve can be individually modeled for every animal. An important aspect of the models for estimating genetic parameters is that they account for the heterogeneity of variance of milk yield on successive days of lactation [17]. This solution increases the accuracy of estimating genetic parameters and breeding values by reducing residual variances. According to Jamrozik et al. [18], the model can be simplified if residual variances is assumed to be fixed-width intervals.

In the overwhelming majority of countries with advanced dairy breeding systems, RRM associated with Legendre orthogonal polynomials is currently used in the official assessment of the genetic value of populations to model the lactation curve [5,6,19,20,21]. One of the problems of using Legendre orthogonal polynomials also implies higher parametrical complexity which may imply determinant coefficient inflation, which is not corrected by the inclusion of logarithmic or exponential terms in the function. However, intensive research continues to address the degree of polynomials used for lactation curve modeling as well as the way observations are grouped according to the heterogeneity of variance [22,23].

Strabel et al. [23] demonstrated that the order of polynomial influences the values of estimated genetic parameters, and that it is appropriate to use fifth-order polynomials to estimate genetic variation in Polish Holstein-Friesian (PHF) cattle. The optimum order of Legendre polynomial was investigated by Khanzadeh et al. [24]. These authors estimated genetic parameters as well as the breeding value for the fat and protein (%) content of milk by modeling the lactation curve using third- to sixth-order Legendre polynomials. They found from their study that the best fit for the additive genetic (AG) and permanent environmental (PE) effects of fat and protein percentages were obtained using the polynomials: 5-AG, 5-PE (fat) and 5-AG, 6-PE (protein). Biassus et al. [6] estimated h^2^ for milk yield (MY), milk protein and fat percentage in Holstein cows using RRM and Legendre polynomials of orders 3 to 6, assuming fixed residual variances during successive days of lactation. It should be highlighted that in this study, genetic variations for milk yield and protein and fat contents, as well as the heritability indicators of these traits followed a similar trend regardless of the order of the polynomial, and the possible differences concerned the beginning and the end of lactation. Similar findings were presented by Costa et al. [15], i.e., variation in AG, PE, h^2^ of milk yield depending on Legendre polynomial as well as the need to use polynomial of 5th order. Moreover, the same authors [15] evaluated the quality of the models and showed the best fit for a model assuming heterogeneity of variance in different lactation periods. Higher order polynomials for modeling the effect of AG and PE than in previous studies were proposed by Bignardi et al. [25] namely seventh- and twelfth-order polynomials, respectively. Contrary results were reported by Naderi at al. [19] for Holstein-Friesian cattle in Iran—out of the different order Legendre polynomials (3rd to 6th), the best fit of the model for the AG and PE effect was obtained for third-order polynomial.

Automation of the milking process, which has been ongoing for around 30 years, as shown by many studies [26,27], may contribute effectively to improving milk yield and quality. At the same time, the automation of milking allows for recording many parameters of the milking process, which are not commonly or directly measured in the conventional system. The extra information includes the interval from the previous milking, number of milkings and cluster attachment time, milking duration, milking speed, milking box time, milk efficiency (milk yield per minute in the milking box), electrical milk conductivity [21,28,29,30,31,32,33,34,35]. It should be highlighted that there are relatively few studies addressing the estimation of genetic parameters for AMS recorded traits [20,32,36].

This study aimed to estimate heritability and genetic correlations for daily milk yield, daily milking frequency and daily milking speed recorded by an automatic milking system throughout the lactation for Polish Holstein-Friesian primiparous.

## 2. Materials and Methods

### 2.1. Data

Data with daily (24-h) records of 1713 primiparous Polish Holstein-Friesian cows from 21 farms with automatic milking systems (AMS) were collected. Herds were equipped with Lely AMS (Astronaut A4). 

Cows were housed in free-stall barns and fed a Partial Mixed Ratio (PMR) feed. The cows received a varied dose of the concentrate, either in the milking robot or the feeding station, depending on the level of their milk yield. Primiparous cows calved from 2011 to 2015 at the age of 18–45 months.

The following data from AMS were chosen for analysis: Milk yield (MY) (kg)—daily milk yield of cow summed during 1 day in milk,Milk frequency (MF) (no.)—number of milking per cow per day,Milking speed (MS) (kg/min)—average milk flow rate during milking (Table 1).

Data on milk performance of primiparous cows milked in AMS was derived from the T4C management and data registration system by Lely East.

Data with outside μ ± 3σ were deleted from the data file. The pedigree file (cows with records and their ancestors) contained 4231 animals in total. Cows were daughters of 702 sires and 1562 dams. Only cows with complete pedigree data were included in the estimation of genetic parameters. Finally, 491,632 records were used for the estimation of (co)variance components.

Two traits RRM were used to estimate the genetic parameters for studied traits. For analysing traits with RRM, Legendre polynomials for the regression on the number of milking day (from test day 5 to test day 305) were used. Two RRM models with first (linear) and second (quadratic) order of fit were used. The residual variance for these two models were homogenous and heterogeneous. The logarithm of the likelihood (logL), Bayesian information criterion (BIC) and Akaike’s Information Criterion (AIC) were used to select the suitable models. The use of RRM in the estimation of genetic parameters makes it possible to determine the heritabilities for individual days of lactation, as well as the genetic correlation coefficients between the daily MF, MY and MS recorded on the same as well on different days of lactation. The obtained results are presented in Figure 1, Figure 2, Figure 3, Figure 4 and Figure 5. The Wombat package [37] was used to estimate parameters. The models with the lowest value of AIC and BIC were the best models.

### 2.2. Statistical Model

In the study we considered using fourth- and fifth-order polynomials, but because convergence could not be reached, we made the estimations with second-order polynomials.

The following model Equation (1) is assumed to be the same for MY, MF and MS:(1)$$y_{\mathrm{ilm}}={\mathrm{Herd}}_{i}+\sum_{k=1}^{n}b_{\mathrm{mk}}{\;z}_{\mathrm{lmk}}+\sum_{k=1}^{n}a_{\mathrm{lk}}{\;z}_{\mathrm{lmk}}+\sum_{k=1}^{n}{\mathrm{pe}}_{\mathrm{lk}}{\;z}_{\mathrm{lmk}}+e_{\mathrm{ilm}}$$
where $y_{\mathrm{ilm}}$ = milking day record m of cow l obtained in herd i, ${\mathrm{Herd}}_{i}$ = fixed effect of herd i, $b_{\mathrm{mk}}$ = fixed regression coefficient specific to days in milk m, $a_{\mathrm{lk}}$ random regression coefficients for additive genetic effect, ${\mathrm{pe}}_{\mathrm{lk}}$ random regression coefficient for permanent environmental, ${\;z}_{\mathrm{lmk}}$ are Legendre polynomials on DIM,  n represent the order of fit and $e_{\mathrm{ilm}}$ is residual effect for each observation and $z_{\mathrm{lmk}}\;$ covariables of Legendre polynomials for the standardized value of the milking day records. 

The model assumptions were expressed by Equation (2):(2)$$E\left[\begin{array}{c} b\\ a\\ pe\\ e \end{array}\right]=\left[\begin{array}{c} b\\ 0\\ 0\\ 0 \end{array}\right],\;v\;(a)=K_{a}⊗A,\;v\;(\mathrm{pe})=K_{\mathrm{pe}}⊗I_{\mathrm{nr}}\;\mathrm{and}v\;(e)\;=\;R$$
with $K_{a}$ and $\;K_{\mathrm{pe}}$ = matrices of (co)variance between random regression coefficients for additive genetic and permanent environmental effects, respectively;  A = additive relationship matrix; $\;I_{\mathrm{nr}}$ = identity matrix; nr = number of animals with records; ⊗ = Kronecker product, and, R = diagonal matrix with a homogeneous residual variance on the diagonal for homogeneous. To fit heterogeneous residual variances model residual covariances differed across 6 stages in each lactation days in milk (DIM): 6 to 50 DIM, 51 to 100 DIM, 101 to 150 DIM, 151 to 200 DIM, 201 to 250 DIM, and 251 to 305 DIM. The number of DIM varied from 64 to 300.

## 3. Results

Random regression models with the order of fit higher than 2 for single-trait analysis and for two trait random regression analysis with the order of fit higher than 1 were not converged due to convergence problems. For MY and MF random regression model with order 2 with heterogeneous residual variance and for MS random regression with order 1 with heterogeneous residual variance was the best model (Table 2). Therefore, the result for MY and MF were reported based on order 2 and for MS based on order 1 was reported (Table 3).

For two traits RRM, based on the model’s quality measures (LogL, AIC, BIC), one can conclude that for MY-MF and MY-MS random regression with order 2 with heterogeneous residual variance and for MF-MS random regression model with order 2 with homogenous residual variance was the best model (Table 4). The result of estimates of additive genetic covariance between random regressions coefficients with the best models were presented in Table 5.

Figure 1 and Figure 2 present changes in genetic variance (AG) and permanent environment (PE) for MY, MF and MS over 305 days of lactation. The curve showing a change of genetic variance MY (AG_MY) is similar to an inverted parabola with higher values in mid-lactation. In turn, the curve showing variance changes for permanent environmental effects MY (PE_MY) has the opposite shape, resembling a parabola with higher values in early and late lactation, and lower values in mid-lactation.

Our study demonstrated that the curves showing changes of genetic variance and permanent environment effects MF were similar in shape to the initial downward trend up to around 50–60 days of lactation, followed by a mildly upward trend. The essential difference between the shape of the two curves was that there was an upward trend for PE variance further into lactation (>230 days) and a downward trend for AG. A markedly different shape concerning to those described earlier for MY and MF, was represented by the curves showing changes of genetic variance and permanent environment effects for milking speed. There was an upward trend over 305-day lactation for AG and a downward trend for PE. 

Daily heritability indicators of the recorded traits showed high variation during lactation, ranging from 0.131 to 0.345 for MY, from 0.153 to 0.322 for MF, and from 0.336 to 0.493 for MS (Figure 3). It must be stressed that the irregular progression in Figure 3 results from the fact that the model accounted for different residual variances in the 305-day lactation periods.

When analysing the shape of the curve showing the values of heritability indicators MY, we observed a moderate downward trend from 5 (0.133) to 21 days of lactation (0.131), and an upward trend to day 160, when heritability reached the maximum value of 0.345. The heritability indicator (0.345) retained its maximum value until 169 days of lactation, after which heritability decreased steadily until the end of lactation (0.212).

We found that MF heritability was highest (0.322) during the early stage of lactation. It decreased to 0.225 over the next 50 days of lactation (until day 55) and increased to 0.227 on day 156 of lactation. The curve showed a downward trend from day 166 to the end of lactation. Regarding daily MS heritability, there was a consistently upward trend (from 0.336 to 0.493) throughout the lactation period (Figure 3). 

The trends shown by the curves (Figure 3) are quantitatively confirmed by averaged daily heritabilities of the recorded traits for the whole lactation, as presented in Table 6. Our study revealed the highest averaged heritability for MS (0.420), followed by MY (0.257) and MF (0.230).

Figure 4 depicts additive genetic (AG) covariances for the pairs of traits MF-MY, MF-MS and MS-MY, which were measured on the same days of lactation. The constructed MS-MY curve resembles an inverted parabola with higher values in mid-lactation. In turn, the curve showing genetic covariance of MF-MY traits has the shape of a parabola with higher values in early and late lactation, and lower values in mid-lactation. Our study showed that genetic covariance for MF-MS was essentially similar throughout the 305-day lactation, with slightly lower values observed in the early and late lactation compared to the mid-lactation. Table 7, Table 8 and Table 9 present genetic correlations calculated from (co)variance components, between daily MF, MY, and MS at different stages of lactation, while Figure 5 those for the same days of lactation.

The genetic correlations estimated between daily MF and MY recorded on different days of lactation ranged from (−0.461) to 0.929. Most of the genetic correlations assumed positive values (Table 7). The strongest genetic relationships between MF and MY (from 0.561 to 0.929) were noted when the results of both traits came from the same days of lactation (Figure 5)—0.705 on average (mean from daily genetic correlations). At the same time, it was observed that the curve showing genetic correlations between MF and MY measured on the same days of lactation tended to assume clearly higher values in the final stage of lactation (Figure 5). It was also observed that the magnitude of these correlations decreased with increasing distance between the days on which MF and MY were recorded (Table 7). In an extreme case, where MF and MY correlated on the extreme days of lactation, the estimated genetic correlations assumed negative values.

The genetic correlations between MF and MS, recorded on different days of lactation, ranged from −0.272 to 0.362. Thus, these estimates are indicative of a weak to moderate correlation between these traits, which in most cases was negative (Table 8). Positive, moderate (>0.3) genetic relationships were observed between MS on around 100 days and MS during the last 60 days of lactation (Table 8).

The curve for the daily correlations between MF and MS, measured on the same days of lactation, showed an upward trend from the beginning to around 180 days of lactation, followed by a downward trend (Figure 5). The estimated correlations ranged from −0.255 to 0.090 (−0.054 on average). Positive relationships between MF and MS were observed in mid-lactation, i.e., from days 123 to 228.

The genetic correlations between MS and MY measured on different days of lactation varied between −0.213 and 0.355, being mostly indicative of a negative relationship (Table 9). The strongest, positive correlations were observed between MS measured beyond 100 days of lactation and MY recorded during the first 100 days of lactation. At the same time, it was noticed that these relationships declined with increasing distance between MS and MY measurement days, after which they gradually changed their sign (direction) from positive to negative (Table 9). Analysis of the daily genetic correlations between MS and MY on the same days of lactation showed that they ranged narrowly from −0.174 to 0.020, averaging −0.057 (Table 9, Figure 5). The curve for the correlation values up to 142 days of lactation showed a mild upward trend and then a downward trend (Figure 5). Positive values of the correlation coefficients between MS and MY were recorded from 104 to 172 days of lactation.

## 4. Discussion

In terms of MY, the study group of Polish Holstein-Friesian cows represented a typical level for the milk recorded cows of this breed [4]. The average levels of MF, MS, and MY in our study fell within the ranges reported by Piwczyński et al. [38] for a population of automatically milked cows in selected European countries and in the USA, namely: MF: 2.50–2.81 milkings/24 h, MY: 22.22–34.07 kg/min, MS: 2.05–2.83 kg/min.

According to Costa et al. [15], it is appropriate to use estimation models with fewer parameters. This derives from the fact that the use of TD results multiplies (ten-fold on average) the amount of data needed to estimate the genetic parameters and breeding value compared to the lactation yield models. This is paralleled by a rapid increase in the number of equations in the system of mixed-model equations necessary for estimating the genetic parameters and breeding value [39]. This is even more justified when estimating the genetic parameters of milk yield traits based on AMS data, which provides information on each milking.

Finally, our study compared the use of first- and second-order polynomials to model variation of MY, MF, and MS. Based on the obtained AIC and BIC values, the best fit was found for the models using second-order polynomials, for both the additive genetic and permanent environmental effects, which additionally accounted for different residual variances at different stages of lactation.

The curve showing the daily heritabilities of milk yield during lactation has the shape of an inverted parabola, which coincides with the results of other authors [8,15,32]. However, many studies reported that the parameters of milk yield were higher in the initial and final stages of lactation than in mid-lactation [9,20,24]. Jamrozik et al. [40] and Strabel et al. [23] consider that this trend is especially evident when assuming the fixed effect of PE for each day of lactation, and, unlike in our study, they assumed a random effect of PE. This course of action is supported by the results of Cobuci et al. [15], who compared the curves of daily h^2^ obtained using two RRM models differing in the mode of PE treatment (fixed or random effect). The model accounting for PE as a fixed effect during the entire lactation period resulted in high h^2^ values in early lactation unlike the model with random PE. Literature on the subject provides several studies which present a completely different shape of the curve for h^2^ values of milk yield per lactation. Kheirabadi [7] and Cobuci et al. [41] presented a curve that showed an upward trend throughout the lactation for h^2^ of milk yield. Yet another trend for h^2^ of MY during successive days of lactation was shown by Bignardi et al. [25] and Nixon et al. [20]. The curves from these studies, showing the values of these heritability values, at first showed a downward trend, followed twice by an upward and a downward trend until the end of lactation. A very similar shape of the curve for daily MY heritabilities, compared to Bignardi et al. [25] and Nixon et al. [20] was obtained by Naderi [19]. The only difference was that there was no downward trend in the initial shape of the curve. Our daily MY heritabilities ranged from 0.131 to 0.345. A similar range of fluctuations for daily heritabilities during lactation to ours, estimated based on TD, was obtained by Biassus et al. [6] (0.14–0.31) and Cobuci et al. [41] (0.15–0.31), and different ranges by Strabel and Misztal [23] (0.14–0.19), Costa et al. [41] (0.27–0.42), Jamrozik and Schaffer [9] (0.40–0.59), Naderi [19] (0.45–0.60) and Moretti et al. [42] (0.14–0.53). In the study by Nixon et al. [20] using 24-h AMS data, the range of daily h^2^ was narrower (0.14 to 0.20) than in our study. In the context of these results, it is necessary to highlight the results obtained by Piwczyński, Sitkowska and Ptak [32] in AMS herds for MY heritability estimated only from the test-day data. The MY heritabilities in this study ranged from 0.162 to 0.338, which is in strict compliance with the range presented here. The averaged MY heritability, calculated from 300 daily indicators (0.257), falls within the ranges reported by other authors (0.12–0.34): Gray et al. [43], Nixon et al. [20], Kirsanova et al. [44], Sasaki et al. [8] and Kheirabadi [7].

Interesting results of studies on genetic variation of AMS recorded traits were reported by Santos et al. [30] based on AMS data from German HF herds. The authors observed that heritabilities differed markedly depending on udder quarter: from 0.05 (left front) to 0.19 (right front). The average MY heritability was 0.10.

Brzozowski et al. [27] and Piwczyński, Brzozowski and Sitkowska [45] demonstrated that changing the milking system from conventional to AMS has a positive effect on increasing the milk yield. According to de Koning, Slaghuis and van der Vorst [46] and Österman et al. [47], this results from increased milking frequency per day. It is therefore important to determine the genetic background of this trait, especially since there are relatively few studies in this area [20,30,32,48]. Carlström et al. [49] estimated that MF heritabilities for Swedish Holstein-Friesian and Swedish Red cows for the first lactation and the second and third lactations together, ranged from 0.02 to 0.07. In turn, König et al. [48] estimated MF heritability for three consecutive 100-day lactation periods to be low: 0.16, 0.19, and 0.22, respectively. In our study, the average heritability calculated from daily values was 0.230, which means that cows can be effectively selected for increased milking frequency. In turn, low heritability (0.05–0.08) of MF was reported by Santos et al. [30].

Nixon et al. [20] estimated daily MF heritabilities based on 24-h AMS data. The constructed curve showed an initial downward trend, followed by an upward trend, a downward trend after mid-lactation, and an upward trend in the last month of lactation. To a certain extent, the shape of the curve coincides with our curve. The difference concerns the much lower range of 24-h heritabilities estimated by Nixon et al. [20] than in our study, which was in the range of 0.02–0.08 vs. 0.153–0.322. A broader range of daily MF heritabilities than in our study was reported by Piwczyński, Sitkowska and Ptak [32] (0.156–0.444). Our study revealed a relatively high MF heritability during the early stage of lactation, which had a direct impact on the range of variation of this indicator during lactation. Strabel and Misztal [50] suggest that the relatively high heritability is due to the small number of test-day milkings that are used to estimate the (co)variance components in these periods and to the fact that the then performed test-day milkings provide the least information. Pool and Meuwissen [51] suggested that the use of milkings only from completed lactations reduces differences between mid-lactation and early and late lactation. Jamrozik and Schaeffer [9] justify the high heritabilities for milk yield and composition (fat, protein) in the early lactation by the importance of this period for calf survival. The authors argue that there is a strict relationship between the quantity and quality of milk, including colostrum that contains antibodies that protect against disease in the first days after calving, and calf survival.

From the viewpoint of production profitability in AMS barns, milk output from the AMS per unit of time is a key factor [52]. For this reason, fast-milking cows are particularly desirable in robotic barns. Research to date has shown relatively high differences in the coefficient of MS heritability (0.14–0.55), which may stem from the way a trait is treated (threshold vs. continuous) and the recording frequency per lactation (recorded for every milking, daily average, or resulting from TD milkings) or in successive lactations.

Research to determine the genetic background (heritability) of milking speed in Canadian Holstein cattle expressed on a 5-point scale was conducted by Sewale, Miglior and Kistermaker [53], who obtained low values depending on the applied model (single, bivariate): 0.14 and 0.1429.

In turn, Berry at al. [28] estimated that h^2^ of average milk flow (kg/min) for Irish Holstein herds using test-day records was 0.21. Gäde et al. [35], Gäde et al. [54], Wethal and Heringstad [31], Carlström at al. [49] and Santos et al. [30] made the estimations in German, Norwegian and Swedish barns with automatic milking. Heritability estimates were based on single milking or 24-h milk yields and ranged from 0.25 to 0.55.

In the studies performed by the authors, the coefficients of heritability were obtained using random regression models (RRM). This modeling method was employed by Amin [14] for estimating the heritability of milk flow in Hungarian Holstein-Friesian cows based on test-day records. The values estimated during lactation ranged from 0.02 to 0.50 (0.20 on average). The curve for heritability changes showed a downward trend in the early lactation (up to 14–16 weeks), followed by an upward trend until the end of lactation. In the study by Piwczyński et al. [32], who also used RRM, daily heritability of MS ranged from 0.252 to 0.665, while the posterior means of heritabilities for 305-d lactation were 0.431. It should be underlined that the curve for changing daily heritabilities in this study assumed relatively high values in the first and last months of lactation. During the remaining period, the curve tended to increase up to 170–180 days of lactation, after which it decreased until 250 days.

In our study, the results pointed to moderate heritability of MS (0.336–0.493), which provides a basis for efficient selection of this trait, and thus for making the farm production more profitable—resulting, inter alia, from the possibility of increasing the number of cows per milking robot. Our findings fall in the upper range of estimates presented by the authors cited above for MS heritability. Of course, profitability of milk production is also affected by other factors, which can be further investigated in subsequent studies.

In the studies performed by the authors, RRM was used to estimate genetic correlations. This allowed for determining the coefficients of genetic correlations between the traits recorded on different days of lactation, as illustrated on surface charts proposed for this type of analysis by Kheirabadi [7]. At the same time, linear graphs were used to illustrate daily coefficients of genetic correlations between correlated traits from the same day of lactation.

As reported by Tse et al. [55], higher yields of the cows milked in AMS barns compared to conventionally milked cows are due to the fact they have free access to the milking robot. In our study, we found generally positive, moderate or even high genetic relationships between MF and MY measured on different days of lactation. This holds in particular for daily coefficients of correlation (0.561–0.929) between MY and MF recorded on the same days of lactation. Santos et al. [30] stressed that high-yielding cows visit AMS more often per day (genetic correlation = 0.49).

Our results are supported by König et al. [48], who estimated the coefficients of genetic correlation between MY and MF for successive 100-day periods of lactation to be 0.47–0.57, 0.46–0.48, and 0.49–0.53, respectively. Differences in the coefficients of genetic correlation in that study depended on the statistical model used. Also consistent with ours are the values of daily coefficients of genetic correlation, reported by Nixon et al. [20] to range from 0.27 to 0.80. It should be underlined that the above authors observed the strongest genetic relationships between MF and MY in the final stage of lactation, which is strictly consistent with our findings Alongside estimating the parameters based on single milking, Nixon et al. [20] determined genetic correlations between MF and MY from TD results. The average coefficient of genetic correlation was 0.14.

In the study by Nixon et al. [20], genetic correlations between daily (24-h) MY and daily (24-h) MF were highest at the end of lactation (0.80) and lowest in mid-lactation (0.27), which concurs with our findings. In the studies conducted by the authors, negative coefficients of genetic correlations were also observed between MF and MY, but this concerned measurements of the correlated traits from opposite periods of lactation. König et al. [48] observed that MF heritability is 0.18, which they considered sufficient for selecting cows against infrequent milkings per day.

Wethal and Heringstad [31] analyzed the relationship between MF and MS based on daily yields recorded by AMS. Their coefficient of genetic correlation (0.14), as in our study (−0.054), shows a weak relationship between these traits. However, in our study, the coefficients generally assumed negative values, which suggests that selection for increased daily milking frequency may contribute to a slight genetic deterioration in milking speed, especially up to 60 days of lactation (rG < (−0.138)). According to the authors, the difference between our estimates and those of Wethal and Heringstad (2019) may result from the applied statistical model. In our study, we used RRM associated with Legendre polynomials, while the authors cited above included the effect of test day and the effect of herd-test day (HTD) as fixed effects.

We concluded from our study that the selection of cows for increased MY in the initial stage of lactation has a positive effect on improving the genetic value of primiparous cows in terms of MS beyond 100 days of lactation. However, the relevant literature revealed no studies exploring the relationship between MY and MS on different days of lactation. Our estimates for the genetic correlations between MY and MS measured on the same days of lactation show weak, mostly negative relationships between these traits. In turn, the average coefficient of genetic correlation (−0.057), based on daily estimates from the whole lactation, indicates there is no relationship between MY and MS.

Our results for the genetic correlations between MY and MS are contradictory to those reported by other authors. Santos et al. [56] estimated genetic correlations using sire models between MY and MS to range widely between 0.36 and 0.73. Such a large variation in the estimates results from the application of different statistical models: recursive linear, linear, linear with regression. In the next study, Santos et al. [30], based on the data collected over 30 days, estimated the coefficient of correlation between MY and MS to be 0.40. Berry et al. [28] estimated the coefficient of genetic correlations between MY and average milk flow (AMF) based on TD results to be 0.69. Gäde et al. [35] estimated the coefficients of genetic correlation between MY and AMF, recorded based on single milking and daily yields in AMS barns, to be 0.51 in both cases. An even stronger relationship between MY and MS was reported by Amin [14]—the coefficients of genetic correlation during lactation ranged from 0.83 to 0.93, while the coefficient for the whole lactation was 0.94. In turn, the curve for daily coefficients of genetic correlations from the beginning to the 8th week of lactation showed a downward trend, followed by an upward trend up to the 28th week, after which it reached a plateau above 0.94.

## 5. Conclusions

The estimated heritabilities for daily milking speed were moderate, while and for daily milking frequency and milk yield were low, which makes it possible to carry out an effective selection, in particular for the first trait. It is known that heritability decreases with increasing breeding pressure. The fact that the genetic variance of milking speed is higher than the milk yield may also provide an advantage in terms of the sustainability of breeding. Considering the high, positive genetic correlation between daily milking frequency and milk yield, it is concluded that giving preference to breeding cows with a natural propensity for making frequent visits to the milking robot, should indirectly improve the genetic base of milk yield.

## Figures and Tables

**Figure 1 animals-11-01943-f001:**
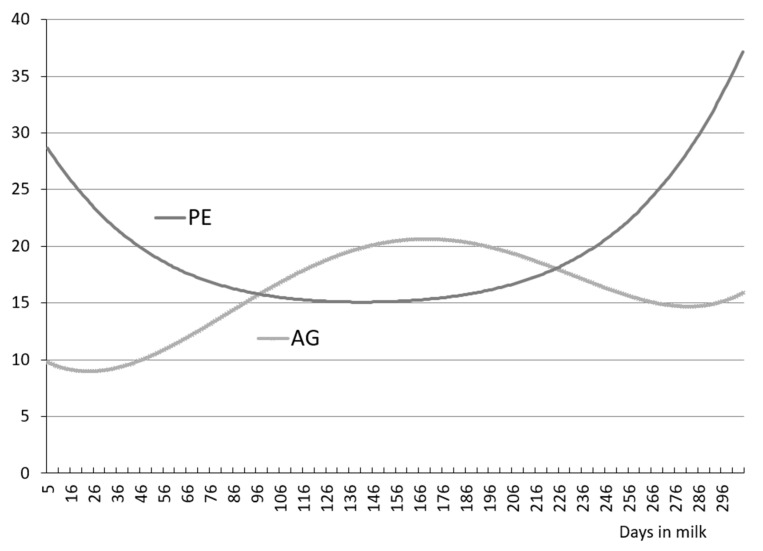
Genetic (AG), permanent environmental (PE) variances for milk yield of primiparous cows.

**Figure 2 animals-11-01943-f002:**
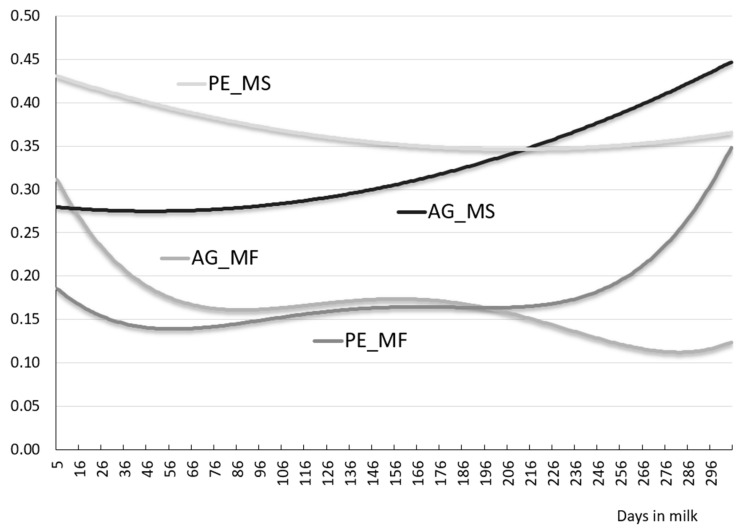
Genetic (AG), permanent environmental (PE) variances for milking frequency (MF) and milking speed (MS) of primiparous cows.

**Figure 3 animals-11-01943-f003:**
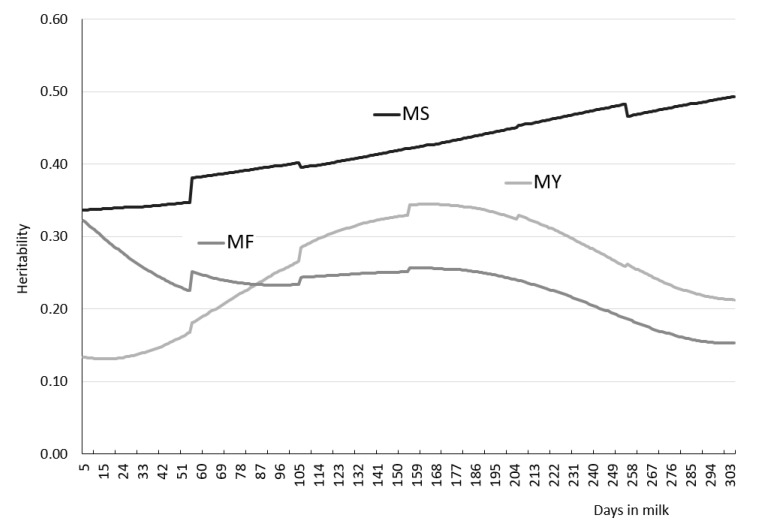
Heritabilities of milk yield (MY), milking frequency (MF) and speed (MS) in subsequent days in milk.

**Figure 4 animals-11-01943-f004:**
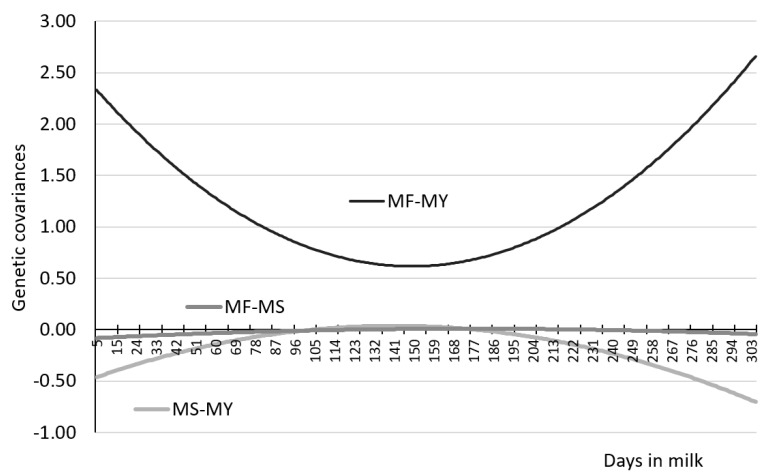
Genetic covariances for controlled pairs of traits primiparous cows, where MS-MY—milking speed and milk yield covariance, MF-MY—milking frequency and milk yield covariance, MF-MS—milking frequency and milking speed covariance.

**Figure 5 animals-11-01943-f005:**
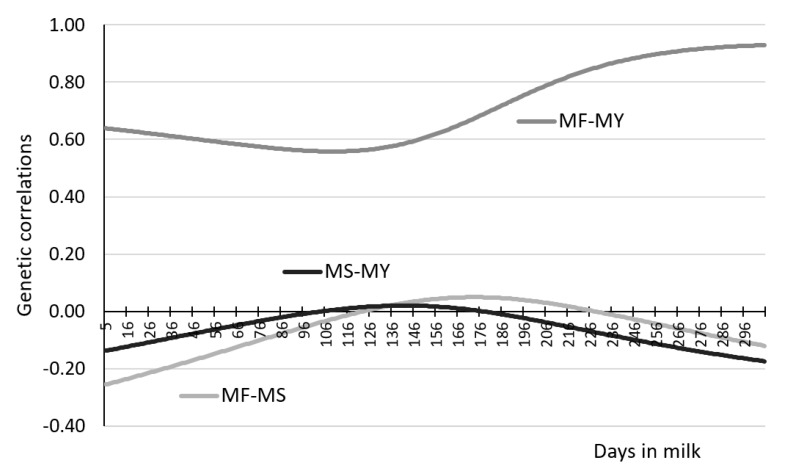
Genetic correlations (rG) for controlled pairs of traits primiparous cows, where MS-MY—milking speed and milk yield covariance, MF-MY—milking frequency and milk yield covariance, MF-MS—milking frequency and milking speed covariance.

**Table 1 animals-11-01943-t001:** Descriptive statistics of primiparous cows-milked in automated milking systems traits.

Trait	Number of 24-h Records	$\bar{x}$	Standard Deviation	Coefficient of Variation (%)
Milk yield (kg)	538,688	28.591	8.823	30.859
Milking frequency (no.)	538,688	2.915	0.886	30.420
Milking speed (kg/min)	538,688	2.526	0.908	35.931

**Table 2 animals-11-01943-t002:** Number of parameters (P), log-likelihood value (Log L), Akaike’s information criterion (AIC) and Bayesian information criterion (BIC) for different models in single trait random regression analysis.

Model	Order of fit	Log L	AIC	BIC	*p*
Milk yield (kg)	1	−1,081,429.608	2,162,883.216	2,163,016.482	12
Milk yield (kg)	2	−1,062,894.572	2,125,825.144	2,126,025.042	18
Milking frequency (no.)	1	−25,033.237	50,090.474	50,223.738	12
Milking frequency (no.)	2	−10,201.294	20,438.588	20,638.484	18
Milking speed (kg/min)	1	382,535.783	765,047.566	764,914.300	12
Milking speed (kg/min)	2	415,176.076	830,316.152	830,116.254	18

**Table 3 animals-11-01943-t003:** Estimates of additive genetic variance (diagonal), covariance (lower diagonal) and correlations (upper diagonal) between random regression coefficient and percentage of variance associated with each eigenvector (EV%).

Trait	Regression Coefficients	Intercept	Linear	Quadratic	EV%
Milk yield (kg)	Intercept	24.13	0.19	−0.73	81.66
	Linear	2.14	4.84	−0.04	14.72
	Quadratic	−5.97	−0.17	2.74	3.63
Milking frequency (no.)	Intercept	0.20	−0.12	−0.53	67.35
	Linear	−0.016	0.086	−0.31	27.73
	Quadratic	−0.043	−0.016	0.032	4.92
Milking speed (kg/min)	Intercept	0.61	0.3144		94.66
	Linear	0.048	0.038		5.34

**Table 4 animals-11-01943-t004:** Log-likelihood value (Log L), Akaike’s information criterion (AIC) and Bayesian information criterion (BIC), number of parameters (P), for different models in two-trait trait random regression analysis.

Model ^1^	Order of Fit ^2^	Log L	AIC	BIC	*p*
MY-MF	1	−1,036,614.637	2,073,247.274	2,073,353.46	9
MY-MF	2 HOM	−923,334.831	1,846,715.662	1,846,987.03	23
MY-MF	2 HET	−907,256.912	1,814,589.824	1,815,038.17	38
MY-MS	1	−909,037.893	1,818,093.786	1,818,199.972	9
MY-MS	2 HOM	−728,555.075	1,457,156.15	1,457,427.518	23
MY-MS	2 HET	−711,685.300	1,423,446.6	1,423,894.946	38
MF-MS	1	−671,185.362	1,342,388.724	1,342,494.91	9
MF-MS	2 HOM	337,838.394	675,630.788	675,359.422	23
MF-MS	2 HET	348,202.475	696,328.95	695,880.604	38

^1^ MY—Milk yield (kg); MF—Milking frequency (no.); MS—Milking speed (kg/min); ^2^ HOM: homogeneous residual variance; HET: heterogeneous residual variance.

**Table 5 animals-11-01943-t005:** Estimates of additive genetic covariance between random regression coefficients in two-trait random regression analysis.

		Milking Speed (kg/min)		Milk Yield (kg)	
		Intercept	Linear	Intercept	Linear
Milking frequency (no.)	intercept	0.015	0.055	1.240	0.210
	linear	−0.033	−0.047	−0.020	1.250
Milk yield (kg)	intercept	0.063	−0.610		
	linear	0.470	−0.410		

**Table 6 animals-11-01943-t006:** Mean daily heritabilities (h^2^) through the whole 305-d lactation.

Trait	AG	SE	PE	SE	R	SE	P	SE	h^2^	SE
MY	15.853	3.091	19.681	2.752	26.789	0.135	62.323	1.311	0.257	0.047
MF	0.164	0.030	0.175	0.027	0.369	0.002	0.709	0.013	0.230	0.041
MS	0.325	0.061	0.367	0.054	0.078	0.000	0.770	0.025	0.420	0.074

Where: MY—Milk yield (kg); MF—Milking frequency (no.); MS—Milking speed (kg/min); AG—additive genetic variances; PE—permanent environmental variances, R—residual variances, P—phenotypic variances, SE—standard error of AG, PE, R, P variances and h^2.^

**Table 7 animals-11-01943-t007:** Genetic correlations between milking frequency and milk yield in different days of lactation.

		Milk Yield (kg) on Different Days of Lactation										
	Days	5	30	60	90	120	150	180	210	240	270	305
**Milking frequency on different days of lactation**	5	0.638	0.632	0.613	0.573	0.495	0.364	0.188	0.010	−0.134	−0.237	−0.318
	30	0.619	0.617	0.606	0.577	0.513	0.398	0.236	0.068	−0.071	−0.173	−0.255
	60	0.582	0.587	0.589	0.577	0.536	0.448	0.312	0.161	0.031	−0.068	−0.150
	90	0.519	0.534	0.553	0.564	0.556	0.506	0.407	0.283	0.167	0.074	−0.005
	120	0.416	0.444	0.484	0.526	0.561	0.564	0.518	0.433	0.341	0.260	0.186
	150	0.262	0.304	0.368	0.447	0.532	0.603	0.626	0.597	0.539	0.478	0.416
	180	0.067	0.121	0.208	0.321	0.458	0.597	0.696	0.731	0.717	0.683	0.639
	210	−0.124	−0.062	0.039	0.175	0.349	0.543	0.707	0.800	0.829	0.823	0.801
	240	−0.275	−0.211	−0.104	0.044	0.239	0.467	0.674	0.810	0.873	0.891	0.888
	270	−0.381	−0.317	−0.209	−0.058	0.146	0.391	0.624	0.788	0.875	0.911	0.922
	305	−0.461	−0.399	−0.294	−0.144	0.063	0.317	0.567	0.751	0.855	0.906	0.929

**Table 8 animals-11-01943-t008:** Genetic correlations between milking frequency and milk speed in different days of lactation.

		Milk Speed (kg/min) on Different Days of Lactation										
	Days	5	30	60	90	120	150	180	210	240	270	305
**Milking frequency on different days of lactation**	5	−0.255	−0.199	−0.126	−0.050	0.025	0.096	0.162	0.221	0.273	0.318	0.362
	30	−0.261	−0.205	−0.131	−0.054	0.021	0.093	0.160	0.219	0.272	0.317	0.362
	60	−0.268	−0.212	−0.138	−0.061	0.015	0.088	0.154	0.214	0.267	0.313	0.359
	90	−0.272	−0.217	−0.144	−0.068	0.006	0.078	0.144	0.203	0.255	0.301	0.346
	120	−0.267	−0.215	−0.147	−0.076	−0.005	0.062	0.124	0.181	0.230	0.273	0.317
	150	−0.244	−0.200	−0.141	−0.080	−0.020	0.038	0.092	0.141	0.184	0.222	0.260
	180	−0.198	−0.166	−0.124	−0.079	−0.034	0.009	0.049	0.085	0.118	0.147	0.175
	210	−0.139	−0.121	−0.097	−0.072	−0.046	−0.020	0.004	0.025	0.045	0.063	0.080
	240	−0.081	−0.076	−0.069	−0.061	−0.052	−0.043	−0.034	−0.026	−0.018	−0.011	−0.003
	270	−0.034	−0.039	−0.045	−0.051	−0.055	−0.059	−0.062	−0.064	−0.066	−0.066	−0.067
	305	0.006	−0.007	−0.024	−0.041	−0.057	−0.071	−0.084	−0.095	−0.104	−0.112	−0.120

**Table 9 animals-11-01943-t009:** Genetic correlations between milk speed and milk yield in different days of lactation.

		Milk Yield (kg) on Different Days of Lactation										
	Days	5	30	60	90	120	150	180	210	240	270	305
**Milking speed on different days of lactation (kg/min)**	5	−0.137	−0.149	−0.166	−0.186	−0.204	−0.213	−0.204	−0.178	−0.148	−0.120	−0.094
	30	−0.090	−0.101	−0.119	−0.140	−0.161	−0.177	−0.179	−0.166	−0.146	−0.126	−0.107
	60	−0.028	−0.039	−0.056	−0.078	−0.104	−0.129	−0.145	−0.148	−0.142	−0.133	−0.122
	90	0.034	0.024	0.007	−0.015	−0.045	−0.079	−0.109	−0.128	−0.136	−0.137	−0.135
	120	0.095	0.085	0.069	0.046	0.013	−0.029	−0.072	−0.107	−0.128	−0.140	−0.147
	150	0.151	0.143	0.128	0.105	0.069	0.019	−0.037	−0.085	−0.119	−0.141	−0.156
	180	0.203	0.196	0.182	0.158	0.120	0.064	−0.003	−0.064	−0.110	−0.140	−0.163
	210	0.248	0.242	0.229	0.206	0.166	0.105	0.029	−0.044	−0.100	−0.138	−0.168
	240	0.288	0.283	0.271	0.248	0.207	0.141	0.057	−0.026	−0.090	−0.136	−0.171
	270	0.322	0.317	0.306	0.284	0.242	0.173	0.082	−0.009	−0.081	−0.133	−0.173
	305	0.355	0.351	0.341	0.319	0.277	0.204	0.107	0.009	−0.071	−0.128	−0.174

## Data Availability

Data is contained within the article.

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
