# Peer review of "Genetic Parameters Estimation of Milking Traits in Polish Holstein-Friesians Based on Automatic Milking System Data"

_animals, 2021, doi:10.3390/ani11071943_

Round 1

Reviewer 1 Report

The manuscript presents an interesting topic and is overall well-written. However, I do not really know if I like the paper as it is. I understand the approach using Legendre polynomials; however, the authors may agree that deep discussion has been raised in the matter and there are as many sources claiming their benefits as those claiming the contrary. Legendre orthogonal polynomials may also imply a higher complexity which implies determinant coefficient inflation, this must be addressed. Still, this does not make the approach here invalid. Certain misconceptions may be corrected such as those comments leading to the misconception of PE being considered fixed. That is, the same factor can be considered (with justification) random (permamnet environmental) or fixed, but if it PE it is always random.

Line 49, I do not know if I totally agree with this sentence “…more efficient selection due to higher h2 values”…low h2 values may also imply a strong selection being carried along the years, hence, the trait variability reduces and predominant phenotypes and the underlying genetic basis fixes in the population.

Line 58. Change former to first. I think it is clearer, as you previously refer to two models.

Line 69. As I said. Legendre orthogonal polynomials also imply higher parametrical complexity which may imply determinant coefficient inflation, which is not corrected by the inclusion of logarithmic or exponential terms in the function.

Line 77. Add author name to citation [24].

Line 310. This is wrong. PE per definition is always a random effect. Another thing would be that you consider treatment or other effect (previously considered pe and thus random) as a fixed effect, still in this case it is fixed and environmental fix effect not a permanent environmental effect. This must be corrected across the manuscript as it shows a flawed conception which may lead readers to confusion.

I do not like the conclusions as they are written. First 4 lines are too general and although results in the paper may support them, this may need to be better clarified here, what is in the body text which supports these conclusions. Lines 472 to 474 may be a good ending sentence for discussion but it is not a conclusion of the work itself. Please rewrite this section as to enable readers to have a message to take home after reading the paper. Remember conclussions must be supported and derive from this paper results and authors must avoid using or suggesting facts that they cannot prove by their results.

Author Response

Thank you very much for the review of our work. We inform that the article was analyzed by a native speaker too.

I understand the approach using Legendre polynomials; however, the authors may agree that deep discussion has been raised in the matter and there are as many sources claiming their benefits as those claiming the contrary. Legendre orthogonal polynomials may also imply a higher complexity which implies determinant coefficient inflation, this must be addressed. Still, this does not make the approach here invalid. Certain misconceptions may be corrected such as those comments leading to the misconception of PE being considered fixed. That is, the same factor can be considered (with justification) random (permamnet environmental) or fixed, but if it PE it is always random.

AU: We have updated the M&M section of the article.

Line 49, I do not know if I totally agree with this sentence “…more efficient selection due to higher h2 values”…low h2 values may also imply a strong selection being carried along the years, hence, the trait variability reduces and predominant phenotypes and the underlying genetic basis fixes in the population.

AU: When formulating the indicated sentence (Line 49), the authors relied on the values formulated by Strabel and Szwaczkowski [24].

Line 58. Change former to first. I think it is clearer, as you previously refer to two models.

AU: We have corrected the sentence as suggested.

Line 69. As I said. Legendre orthogonal polynomials also imply higher parametrical complexity which may imply determinant coefficient inflation, which is not corrected by the inclusion of logarithmic or exponential terms in the function.

AU: We add the following sentence: „ One of the problem of using Legendre orthogonal polynomials also imply higher parametrical complexity which may imply determinant coefficient inflation, which is not corrected by the inclusion of logarithmic or exponential terms in the function”.

Line 77. Add author name to citation [24].

AU: We have corrected the sentence as suggested.

Line 310. This is wrong. PE per definition is always a random effect. Another thing would be that you consider treatment or other effect (previously considered pe and thus random) as a fixed effect, still in this case it is fixed and environmental fix effect not a permanent environmental effect. This must be corrected across the manuscript as it shows a flawed conception which may lead readers to confusion.

AU: In our analysis, we assumed PE as a random and permanent environmental effect and not necessary to change it to environmental fixed effect.

I do not like the conclusions as they are written. First 4 lines are too general and although results in the paper may support them, this may need to be better clarified here, what is in the body text which supports these conclusions. Lines 472 to 474 may be a good ending sentence for discussion but it is not a conclusion of the work itself. Please rewrite this section as to enable readers to have a message to take home after reading the paper. Remember conclussions must be supported and derive from this paper results and authors must avoid using or suggesting facts that they cannot prove by their results.

AU: The Conclusions section was rewriten.

Reviewer 2 Report

Animals-1245044

This paper aims to estimate heritability and genetic correlations for milk yield, and milking frequency and speed of primiparous Holstein cows milked in AMS. Materials and methods section suffers from a lack of information to let the reader understand what has been done. First, I had some questions as: “From the objective, this is not clear that authors only use one day of data per cow. This is very important. Why only 24 h of data were used? What is the purpose behind that?”, but I think that these questions emerged from a lack of information in the M&M section. The objective is not clear and should be rephrased. 

Authors refer to herd test day, but data were from AMS. Please clarify.  

How the herds were enrolled?

Authors use many abbreviations: BV, HV, RV, RRM, FRM, AG, PE. I suggest to reduce the number of abbreviations that could lost the reader.

Line 26: “milking cow”

Line 69: this abbreviation has been already described.

Line 81: This is not clear from the text what 5, 5 and 5,6 refer to. I think that for protein, 5 is for additive genetic and 6 for permanent environment, but it is not specified.

Line 85: “protein and fat contents”

Lines 91-93: Not clear what the authors mean by this. Is it from a reference or this is their thoughts? I am not sure if this sentence is necessary?

Line 108: Objective: add throughout the lactation?

Line 116: Define PMR at first use. This is not the PMR which is fed individually, but concentrate in the milking robot. Please rephrase.

Line 117: indicate that cows are primiparous.

Line 120: why successive? Why not only number of milking per cow per day?

Table 1: I am not sure to understand the number of 24-h records (538,688) with only 1,713 cows. So, this is repeated measurements from DIM x to xxx? This is not clear from the text. This should be clarified. In the table title, specify that cows were milked in automated milking systems.  

Line 123: How did you delete over 40,000 records? Outliers?

Line 124: ok now 4,231 animals???

Line 126: This sentence should be rephrased. Hard to understand.

Line 127: Why referring to test day? Cows were milked in automated milking systems, right?

Line 128: you used 2 models: both with first and second orders, right? One model with the assumption of homogeneity of variance and the other one with heterogeneity? Not clear as written. More details about the Legendre polynomial needed (equations?). if previous authors obtained that 5 and 6 orders of fit were better, why not using them in your model?

Line 131: should have the lowest value?

Line 135: name the 3 traits.

Line 136: what is the herd test day effects? Random effect of the herd?

Line 150: What HTD stands for?

Table 2: Order of fit 2 and 3, in the statistical section (lines 128-129), it was written linear and second, not third. Not clear.

Figures 1 and 2: Is there a unit for the y axis?

Lin 249: not clear why the authors discuss about data taken at different days? It was not specified in the M&M. what was the purpose behind that? Line 252: different days? Could you explain?

Line 286: replace – by : as MF: 2.50-2.81, etc.

Line 294-295: should be placed in the M&M section.

Line 297: this was not the aim of this paper according to line 108 (model variation of ...).

Line 363: “suggested”

Line 459: this reference is not in the reference section

Author Response

Thank you very much for the review of our work. We inform you the article was analyzed by a native speaker. 

Comments and Suggestions for Authors

Animals-1245044

This paper aims to estimate heritability and genetic correlations for milk yield, and milking frequency and speed of primiparous Holstein cows milked in AMS.

Materials and methods section suffers from a lack of information to let the reader understand what has been done.

First, I had some questions as: “From the objective, this is not clear that authors only use one day of data per cow. This is very important. Why only 24 h of data were used? What is the purpose behind that?”, but I think that these questions emerged from a lack of information in the M&M section.

AU: We have clarified the purpose of the research and we enriched the M&M section with necessary additions.

The objective is not clear and should be rephrased. 

AU: We have clarified the purpose of the research.

Authors refer to herd test day, but data were from AMS. Please clarify.  

AU: We made a mistake introducing the phrase "test day". In fact, we estimated the genetic parameters based on the 24h performance.

How the herds were enrolled?

AU: We inserted the following sentence “Data on milk performance of primiparous cows milked in AMS was derived from the T4C management and data registration system by Lely East.” To M&M section.

Authors use many abbreviations: BV, HV, RV, RRM, FRM, AG, PE. I suggest to reduce the number of abbreviations that could lost the reader.

AU: We appreciate the comment, however, we  decided to leave the abbreviations RRM, AG and PE as they are commonly used in describing the scientific findings on genetic parameters.

Line 26: “milking cow”

Au: Corrected

Line 69: this abbreviation has been already described.

AU: Corrected.

Line 81: This is not clear from the text what 5, 5 and 5,6 refer to. I  . that for protein, 5 is for additive genetic and 6 for permanent environment, but it is not specified.

AU: Corrected.

Line 85: “protein and fat contents”

AU: Corrected.

Lines 91-93: Not clear what the authors mean by this. Is it from a reference or this is their thoughts? I am not sure if this sentence is necessary?

AU: Controversial sentence has been removed.

Line 108: Objective: add throughout the lactation?

AU: Corrected.

Line 116: Define PMR at first use. This is not the PMR which is fed individually, but concentrate in the milking robot. Please rephrase.

AU: The sentence was rephased.

Line 117: indicate that cows are primiparous.

AU: Completed.

Line 120: why successive? Why not only number of milking per cow per day?

AU: We have removed the word “successive”.

Table 1: I am not sure to understand the number of 24-h records (538,688) with only 1,713 cows. So, this is repeated measurements from DIM x to xxx? This is not clear from the text. This should be clarified. In the table title, specify that cows were milked in automated milking systems.

AU: This is exactly how such a large number of records were obtained. The title of Table 1 was corrected as suggested.

Line 123: How did you delete over 40,000 records? Outliers?

AU: Data with outside μ± 3? were deleted from the data file. Finally, 491,632 records.

Line 124: ok now 4,231 animals???

AU: Yes, cows with record and their ancestors.

Line 126: This sentence should be rephrased. Hard to understand.

AU: The sentence was rephrased.

Line 127: Why referring to test day? Cows were milked in automated milking systems, right?

AU: Corrected to “milking day”.

Line 128: you used 2 models: both with first and second orders, right? One model with the assumption of homogeneity of variance and the other one with heterogeneity? Not clear as written. More details about the Legendre polynomial needed (equations?). if previous authors obtained that 5 and 6 orders of fit were better, why not using them in your model?

Line 131: should have the lowest value?

AU: The models with lowest value of AIC and BIC were the best models.

Line 135: name the 3 traits.

AU: Completed.

Line 136: what is the herd test day effects? Random effect of the herd?

AU: The herd effect was fixed. We have included this information in the description of the analysis model.

Line 150: What HTD stands for?

AU: We mislabeled this analysis component. There should only be Herd.

Table 2: Order of fit 2 and 3, in the statistical section (lines 128-129), it was written linear and second, not third. Not clear.

AU: We corrected table 2.

Figures 1 and 2: Is there a unit for the y axis?

AU: We removed y axis descriptions.

Lin 249: not clear why the authors discuss about data taken at different days? It was not specified in the M&M. what was the purpose behind that?

AU: We completed M&m section.

Line 252: different days? Could you explain?

AU: The paper presents the genetic correlation coefficients between the controlled traits recorded on the same as well as on different days of lactation. Thanks to this, it is possible to predict the effects of selection, e.g. on milking speed in the final lactation stage, based on the daily milk yield from the initial stage of lactation.

Line 286: replace – by : as MF: 2.50-2.81, etc.

AU: Corrected.

Line 294-295: should be placed in the M&M section.

AU: Corrected

Line 297: this was not the aim of this paper according to line 108 (model variation of ...).

AU_ We corrected the goal of the article (Line 108).

Line 363: “suggested”

AU: Corrected.

Line 459: this reference is not in the reference section.

AU: We have corrected the reference

Round 2

Reviewer 1 Report

Line 49. What turns efficient when h2 values are high is the response to selection (selection potentiality). Still high values of h2 may rather evidence high variability (a sign of lack of previous selection) than the evidence of efficient selection.

Conclussions still report results rather than factual conclussions derived from such results. This is, for instance instead of the authors saying ….The estimated heritabilities for daily milking speed were moderate, and for daily milking frequency and milk yield were low….which is results, what does this information tells us.

Author Response

Line 49. What turns efficient when h2 values are high is the response to selection (selection potentiality). Still high values of h2 may rather evidence high variability (a sign of lack of previous selection) than the evidence of efficient selection.

AU: The controversial sentence has been removed from the article.

Conclusions still report results rather than factual conclussions derived from such results. This is, for instance instead of the authors saying ….The estimated heritabilities for daily milking speed were moderate, and for daily milking frequency and milk yield were low….which is results, what does this information tells us.

AU: The conclusions have been redrafted.

AU: Please be advised that the text of the article has been re-verified by a native speaker.

Reviewer 2 Report

Thank you for having addressed my comments.

One more: Why herd was not a random effect in the model. This should be justified.

Author Response

One more: Why herd was not a random effect in the model. This should be justified.

AU: Herd considered as fixed effect because we only 21 herd. There is not rule in this matter but in most cases a effects with level more than 50 are considered as a random, e.g.:

https://doi.org/10.3168/jds.2007-0382.

In this study there is only 96 herd and herd_test days effect is considered fixed.